# Research Progress on Porcine Reproductive and Respiratory Syndrome Virus NSP7 Protein

**DOI:** 10.3390/ani13142269

**Published:** 2023-07-11

**Authors:** Huawei Li, Qin Luo, Huiyuan Jing, Yuzhen Song, Weili Kong, Mengmeng Zhao, Qingge Zhu

**Affiliations:** 1College of Food and Bioengineering, Henan University of Animal Husbandry and Economy, Zhengzhou 450047, China; centrosome@126.com; 2Department of Veterinary Medicine, School of Life Science and Engineering, Foshan University, Foshan 528000, China; luoqin121104@163.com; 3College of Veterinary Medicine, Henan University of Animal Husbandry and Economy, Zhengzhou 450047, China; lhsjhy@126.com (H.J.); zzmzsong@163.com (Y.S.); 4Gladstone Institutes of Virology and Immunology, University of California, San Francisco, CA 94158, USA; weili.kong@gladstone.ucsf.edu

**Keywords:** PRRS, PRRSV, NSP7, research progress, biological characteristics

## Abstract

**Simple Summary:**

Research on the NSP7 protein is of great significance in the diagnosis, prevention, and control of porcine reproductive and respiratory syndrome virus (PRRSV). We summarize its genetic variation, recombination hotspots and breakpoints, replication, virulence, immune mechanisms, and interaction with viral proteins and host proteins, which provide theoretical support for its application in PRRS diagnosis, novel vaccine design, and therapeutic drug development.

**Abstract:**

Porcine reproductive and respiratory syndrome (PRRS) is a highly contagious and severe infectious disease caused by the PRRS virus (PRRSV). PRRS is characterized by reproductive disorders in sows and respiratory dysfunction in pigs. Non-structural protein 7 (NSP7) is one of the most conserved functional proteins in PRRSV, and it plays an important role in viral replication and humoral immune responses in infected hosts. This review discusses the biological characteristics of NSP7 to provide theoretical support for its application in PRRS diagnosis, novel vaccine design, and therapeutic drug development.

## 1. Introduction

Porcine reproductive and respiratory syndrome (PRRS), commonly known as “blue-ear pig disease”, is caused by the PRRS virus (PRRSV). PRRS mainly manifests as reproductive disorders in sows and respiratory disease in pigs at various life stages. PRRS was classified as a category B infectious disease by the World Organization for Animal Health (OIE) in 1992 [1]. The common clinical manifestations of PRRS include abortion, stillbirth, infertility, weak piglets in pregnant sows, hind limb weakness, inability to stand, ataxia, high fever, and respiratory diseases in pigs. PRRS is one of the most severe pig diseases that cause significant economic loss to the pig industry.

In 1987, a PRRS outbreak was first recognized in Midwest USA. During the subsequent decades, PRRS spread rapidly in Europe and North America. In 1991, PRRSV was first reported in Taiwan, China, and it was first isolated in China in 1996 and identified as the CH-1a strain [2]. PRRSV is divided into two genetic species, according to differences in gene sequences and countries where the PRRSV species was first isolated: European type 1 and North American type 2, also known as PRRSV-1 and PRRSV-2, respectively. Based on the International Committee on Taxonomy of Viruses (ICTV), PRRSV-1 was classified as the species *Betaarterivirus suid* 1 and PRRSV-2 as *Betaarterivirus suid* 2. The homology of the two species is approximately 60% [3]. At present, PRRSV-2 strains are dominant in China. PRRSV has evolved into various strains with the continuous acquisition of mutations and genomic recombination. For example, in 2006, there was an outbreak of highly pathogenic PRRSV (HP-PRRSV) caused by a mutant PRRSV with a 30-amino-acid deletion in the *NSP2* region. This highly pathogenic strain rapidly invaded many parts of China [4], resulting in significant economic losses to the pig industry. In 2008, the NADC30 strain appeared in the USA. In around 2013, a PRRSV strain with high homology to the American strain NADC30 appeared in China, which was called NADC30-like PRRSV [5], and this epidemic affected 12 provinces. Recently, a new variant strain of PRRSV, PRRSV 1-4-4 lineage 1C [6], was identified in the western USA. The transmission and mortality rates of this new strain are higher than those of previous strains, and the vaccination effect is poor. To date, no specific drugs are available to treat PRRS. The significant differences in gene sequences, heritability, antigenicity, pathogenicity, and other biological characteristics among different PRRSV strains pose great challenges for PRRS diagnosis, as well as vaccine and drug development.

PRRSV is a single-stranded positive RNA virus that belongs to the order *Nidovirus* in the family, *Arterioviridae*. PPRSV mainly infects the respiratory and reproductive systems of animals, destroying the host macrophages, thus evading the host’s immune response. The genome is approximately 15 kb in length, has an undivided envelope, and contains at least 10 open reading frames (ORFs) [7], with ORF1a and ORF1b accounting for approximately 75% of the viral genome. The replicase PP1a, encoded by ORF1a, can be hydrolyzed into 10 NSPs, namely NSP1–NSP8, with NSP1 further hydrolyzed into NSP1α and NSP1β, whereas NSP7 is further hydrolyzed into NSP7α and NSP7β [8]. The *NSP7* gene is located between the *NSP6* and *NSP8* genes. The 3C-like serine protease of PRRSV NSP4 acts on a highly conserved protease cleavage site of NSP7 to cleave the protein into the NSP7α and NSP7β forms, with the former being the more highly conserved PRRSV NSP7. The C-terminus of NSP7α comprises a central proline region, with proline residues making up approximately 10 of the 20 amino acids in the tail, whereas the N-terminus of NSP7β comprises four prolines and five other amino acids [8]. The secondary structure of PRRSV NSP7α is similar to that of the homologous equine arteritis virus (EAV) NSP7α protein. A three-dimensional structural comparison of PRRSV and EAV NSP7α proteins showed the highest degree of conservation in the region spanning α-helix 2 to α-helix 3, whereas PRRSV NSP7α has one more proline-rich region at the C-terminus than EAV NSP7α [8]. In addition, NSP7β is easy to degrade. Chen et al. [9] only detected cleaved NSP7α, whereas cleaved NSP7β was not detected in the infected cells by Western blotting and radioimmunoassay. To further explore the molecular structure of NSP7α, Chen et al. [9] used gene cloning technology to express recombinant proteins with an N-terminal histidine tag and employed nuclear magnetic resonance for structural analysis. The results showed that the C-terminus of NSP7α was irregularly curled and contained an amino acid-rich region, which was more flexible than the other regions.

NSP7 contains 269 amino acids in PRRSV-1 and 259 amino acids in PRRSV-2. The study of gene structure and function is of great significance for the differential diagnosis of different strains. The length of the *NSP7* gene is approximately 777 bp [10], which is highly conserved in different PRRSV strains with no significant sequence divergence noted. Furthermore, NSP7 has high sensitivity and monoclonal antibody specificity. By truncating PRRSV NSP7 and synthesizing polypeptides, the N^145^AWGDEDRLNKKK^157^ region of NSP7 protein was identified as a highly conserved B-cell epitope, which can be recognized by PRRSV-positive serum; this epitope is highly specific to PRRSV and is conserved among different strains [11]. Additionally, NSP7 shows good immunogenicity. Janková et al. [12] showed that the NSP7 antigen proved to be suitable for serological detection of PRRS-specific antibodies and can be used to distinguish post-vaccination and post-infection antibodies in pigs vaccinated with inactivated vaccines. Overall, research on the genetic variation, structural function, and immunogenicity of the *NSP7* gene will be helpful for the design and development of PRRSV antibody detection reagents and new genetic engineering strategies for vaccine development.

This review provides a comprehensive overview of the NSP7 protein in PRRSV, including its genetic variation, recombination hotspots and breakpoints, replication, virulence, immunity, and interactions with viral and host proteins. The aim of this review is to establish a fundamental theoretical basis for the diagnosis and prevention of PRRSV.

## 2. *NSP7* Genetic Variation

The *NSP7* gene sequence is relatively conserved among PRRSV strains. Studies have shown that the homology of the NSP7 protein-coding region between the same strains is relatively high. The PRRSV-1 homology rate can reach up to 96.7–97.4%, and the PRRSV-2 homology rate can reach up to 84.9–100% [13,14]; the homology of NSP7 between different strains is higher than that between NSP1 and NSP2 [15]. The overall homology of NSP7 between PRRSV-1 and PRRSV-2 strains is approximately 45%. In addition, a deletion mutation of the NSP2 amino acid sequence that changes the start and end sites of the *NSP7* gene can cause changes in the biological function of the protein. The start and stop sites of the PRRSV-1 *NSP7* gene are Ser^2083^ and Glu^2351^, respectively, whereas those of PRRSV-2 are Ser^2200^ and Glu^2458^, respectively [16].

Zhang et al. [17] found that a single-site mutation of the PRRSV NSP7α F72 amino acid residue had a fatal effect on PRRSV reactivation in MARC-145 cells. The NSP7β locus at amino acid site 38 is relatively conserved among the arteritis virus families; however, deletion of the conserved NSP7β region and the NSP7β amino acid 37/38 mutation has a fatal effect on viral replication and proliferation, suggesting that the amino acid 37/38 locus is a key site for NSP7β function [18]. In addition, NSP7 mutation in different strains potentially affects the virulence and proliferation of the virus. Li et al. [19] isolated and inoculated two PRRSV strains, GDQJ and GDBY1. Homology and phylogenetic analysis showed numerous variations in the GDQJ genome, distributed in the 5′ untranslated region (UTR), *NSP1β*, *NSP2*, *NSP3*, *NSP5*, *NSP7*, *NSP9*, *NSP10*, *GP5*, and *N* regions. Multiple sequence alignment analyses showed that the PRRSV attenuated strain S1 contained the same amino acid residue (R) as the GDQJ strain at 188 sites in the *NSP7* region, which provided a foundation for further study on the effect of NSP7 on PRRSV virulence. The PRRSV TJ strain was inoculated into MARC-145 cells for continuous subculture to weaken the virus, then inoculated into pigs, and the pathogenicity of PRRSV TJ strain to the pigs was weakened. The genetic variation analysis of different generations of viruses during the weakening process showed that there were different degrees of variation in each gene. When the TJ strain was subcultured to F140, there were 40 amino acid mutations in NSP1-5, NSP7, and NSP9-11 [20]. In addition, studies have shown that PRRSV virulence-related genes are mainly located in NSP3-8 and GP5 [21]. Therefore, it is hypothesized that the genetic variation of NSP7 has an effect on the attenuation of PRRSV virulence.

## 3. Recombination Hotspots and Breakpoints in *NSP7* of Some PRRSV Strains

Various PRRSV strains have recombination hotspots and breakpoints in the *NSP7* region, which are important for studying the genetic variability and virulence of PRRSV. Sequencing of the recombination breakpoints in PRRSV-2 strains of different lineages revealed that recombination hotspots in the *NSP7* region are involved in replication with 296 recombination events, accounting for approximately 4.04% of all recombination events in the *GP2* and *GP4* regions (4.89% and 5.05%). This indicates that only a few regions of the *NSP7* gene involved in replication are prone to recombination [22]. Xu et al. [23] analyzed the recombination of lineage 3 PRRSV QYYZ-like strains that persisted in China and found recombination hotspots in the *NSP2*, *NSP7*, *ORF2a*, and *ORF3* regions. PRRSV genome recombination in these regions may be associated with viral proliferation, cell tropism, and increased virulence, which are conducive to viral survival and spread. In addition, *NSP7* gene breakpoints have been identified in some strains. The recombination phenomenon of 20 PRRSV strains was analyzed using seven algorithms with RDP4 software and SimPot identification. This showed that the SDYT-p13-2013 strain has potential recombination breakpoints in the 6638–7696 nucleotide region of *NSP7* and *NSP9* [24].

## 4. Effect of *NSP7* on Viral Replication

NSP7β restores the activity of virus particles [17] and may be involved in PRRSV proliferation and replication. In addition, the deletion of any fragments of the *NSP7α*, *NSP7β*, or *NSP7* coding regions can inhibit virus production. NSP7 may play an important role in virus replication [25]. NSP7 can affect viral replication and proliferation by interacting with other proteins. When NSP7α, NSP7β and NSP7 deletion infectious clones were constructed and transfected into baby hamster kidney cells for rescue, the results showed that only the complete full-length virus could rescue the active complete virus. This suggests that NSP7α, NSP7β, and NSP7 may play a key role in viral replication and proliferation [25]. An additional study verified that *NSP1*, *NSP2*, *NSP7*, *NSP9*, and 3′-UTR were related to the high replication efficiency of HP-PRRSV by exchanging genes between HP-PRRSV and infectious cloned PRRSV [26]. NSP7 is a long-term intermediate in PRRSV replication, which regulates viral replication and proliferation. Chen et al. [27] produced monoclonal antibodies against NSP7α, and NSP7β expressed in prokaryotic cells and detected their expressions in MARC-145 cells and porcine alveolar macrophages (PAMs) by Western blotting and immunofluorescence. The results showed that NSP7α and precursor NSP7 proteins were the only intermediate viral replication products and participated in physiological responses in the host cytoplasm. NSP5-8 may be the major long-lived intermediate in HP-PRRSV-infected cells. These proteins play an important role in the PRRSV life cycle; however, their specific biological functions require further study [28].

PRRSV mainly infects PAMs and inhibits the innate immune regulation of the host. Interferon regulatory factor 7 (IRF7) regulates the expression of interferon (IFN)-stimulated genes (ISGs). Studies have shown that NSP7 can downregulate IRF7 expression, thereby inhibiting IFN and ISG expression and promoting viral replication and proliferation [29]. A rabbit polyclonal antibody against NSP7 protein was prepared and co-inoculated with PRRSV in MARC-145 cells. Polyclonal antibodies against NSP7 at different dilutions inhibited viral replication [10], and the antibody prepared at 10^−3^ dilution had the most obvious effect. Another study [30] determined that host RNA-binding motif protein 39 (RBM39) can bind to PRRSV RNA (*NSP4*, *NSP5*, *NSP7*, *NSP10-12*, *M*, and *N* genes) and promote PRRSV proliferation by downregulating IFN-β production. An E3 ubiquitin ligase ring finger 122 (RNF122) protein helicase-like transcription factor, which NSP1α and NSP7 downregulate, can inhibit RNF122 promoter activity and promote RNF122 transcription. RNF122 ubiquitinates the pathogen pattern recognition receptor melanoma differentiation-associated gene 5 (MDA5) protein via amino acid residues—K27 and K48, thereby degrading MDA5, inhibiting IFN production, and ultimately promoting virus proliferation [31]. Thus, this study provided new insights into the mechanism of action of NSP7 in viruses.

PRRSV replication produces high titers of NSP-specific antibodies. NSP7 mutations and deletions affect the PRRSV replication cycle, hindering viral mRNA and protein synthesis and inhibiting viral replication and survival. Zhang et al. [17] determined the structure of *NSP7* regions and identified NSP7 mutations and deletions. Most of the NSP7 residue substitutions and deletions reduced the highest titer concentration of the virus and were fatal to its recovery. It was hypothesized that NSP7 affects viral RNA synthesis and key regions of protein translation. This study provided insights into the search for sites of action of target proteins for developing new antiviral drugs and therapeutics. NSP7 promotes viral replication; some proteins can inhibit viral replication by affecting NSP7 expression. Zinc finger antiviral protein (ZAP) is a host restriction factor encoded by the antiviral gene that evolved over the long-term coexistence of the virus and host. ZAP contains two variants, ZAP-L and ZAP-S. ZAP-L can inhibit viral replication by degrading viral proteins and reducing NSP7β expression. Transfecting a PRRSV NSP-expressing eukaryotic expression plasmid and Flag-ZAP-L into HEK-293 T cells showed that ZAP-L inhibited NSP7, NSP9, and NSP12 expression in a dose-dependent manner. ZAP-L may inhibit viral replication by degrading certain viral proteins [32].

## 5. Role of NSP7 in PRRSV Virulence

Studies have shown that NSP3-8 and ORF5 are the main determinants of PRRSV virulence [21]. To determine the mutations associated with the attenuation of the HP-PRRSV TJM vaccine strain, Leng et al. [33] subcultured the HP-PRRSV TJ virus in MARC-145 cells and analyzed its F19, F46, and F78 sequences. It was hypothesized that 31 amino acid variations were distributed in NSP1β, NSP2-5, NSP7, NSP9, NSP10, GP4, and GP5, and the deletion of 120 amino acids in the NSP2 region starting from F19 was suggested as the reason for PRRSV virulence weakening.

Tan et al. [34] isolated a novel HBap4/2018 strain from a recombinant HP-PRRSV-like strain and a NADC30-like strain. The *NSP2*–*NSP3* (nucleotides 2000–5105), *NSP5*–*NSP7* (nucleotides 6292–7412), and *ORF* 3′-*UTR* (nucleotides 14,249–15,003) gene regions from the NADC30-like strain matched most of the genes from the HP-PRRSV-like strain. In 4-week-old piglets infected with the HBap4/2018 strain, the strain remains highly pathogenic and can cause secondary infections in pigs, similar to HuN4 infections. Further analysis showed that the *NSP2*–*NSP3* (nucleotides 2000–5105), *NSP5*–*NSP7* (nucleotides 6292–7412), and *ORF* 3′-*UTR* (nucleotides 14,249–15,003) gene region substitutions did not significantly affect HP-PRRSV virulence. In conclusion, whether NSP7 has an effect on PRRSV virulence is still controversial.

## 6. Role of NSP7 in Escape of PRRSV from Host Immune Mechanisms

By constructing a B-cell tetramer of PRRSV NSP7, Rahe et al. [35] detected rare antigen-specific B cells. The NSP7–B-cell tetramer helps characterize the PRRSV-specific memory B-cell response and further contributes to the study of NSP7 and the understanding of the host humoral response to PRRS. Yan et al. [36] used the online Immune Epitope Database network (http://www.iedb.org/ (accessed on 5 May 2019).) to predict the NSP7 epitope and found multiple B-cell epitopes.

Tumor necrosis factor-alpha (TNF-α) has antiviral properties. NSP7, NSP11, and NSP12 affect the differential expression of TNF-α mRNA induced by different pathogenic strains, thus affecting PRRSV escape from host immunity [37]. Liu et al. [38] infected porcine macrophages with a virulent and attenuated PRRSV strain. They detected differences in IFN-λ expression at the mRNA and protein levels by fluorescence quantitative polymerase chain reaction (PCR) and Western blotting, respectively. The results showed that NSP7α hardly affected the mRNA or protein expression of IFN-λ3, whereas NSP7β and NSP9 co-transfection into PK-15 cells upregulated IFN-λ3 mRNA expression but did not affect the protein level. It is hypothesized that NSP7β and NSP9 inhibit the immune response via IFN-λ3, enabling PRRSV to evade the immune mechanism of the host. IFN-α plays an important role in the acquired immune response of the body and the response to antivirals [39]. NSP7, NSP12, ORF3, and ORF7 can significantly inhibit the expression of IFN-α-stimulated response elements, inhibit IFN-α downstream signaling pathway activation, and downregulate IFN production, thereby inhibiting the host from achieving acquired immunity [40].

PRRSV can infect adjacent cells by connecting nanotubes between cells, thereby avoiding neutralizing host antibody responses and effectively escaping host immune mechanisms [41]. Furthermore, F-actin and myosin of the intercellular nanotubes can co-precipitate with PRRSV NSP1, NSP2, NSP4, NSP7–NSP8, GP5, and N proteins to promote the formation of nanotubes and virus clusters in virus-infected cells [41]. Studying the mechanism of PRRSV escape from host immunity is of great significance for preventing and controlling PRRS.

## 7. NSP7 Interacts with Viral Proteins

PRRSV NSP9 is an RNA-dependent RNA polymerase (RdRp) involved in forming the viral replication and transcription complex (RTC). Using yeast two-hybrid experiments, Chen et al. [8] found that PRRSV NSP7α interacts with the RdRp structure of NSP9, which may play an important role in the viral RTC or assist NSP9 in PRRSV RNA synthesis. In particular, the L69 and F72 amino acid residues of NSP7α play an important role in this process; however, the mechanism of action requires further study. NSP9 and NSP12 are the core components of the RTC [42]. Lan et al. [43] demonstrated that NSP7 can interact with NSP9 and NSP12 to form a transcriptional complex, affecting viral replication and proliferation, while NSP7 interacts with NSP9β and NSP12. In addition, pull-down experiments proved that NSP5 can interact with NSP9 indirectly through NSP7α. However, this study failed to verify the interaction between NSP7β and NSP5, presumably due to the instability of NSP7β [43]. In summary, NSP7 interacts with NSP5, NSP9, and NSP12 and participates in viral transcription and proliferation. The specific interaction between NSP7 and viral proteins is shown in Figure 1. The interaction between NSP7 and viral proteins affects viral replication and activity, which provides a new idea for the development of anti-PRRSV drugs.

## 8. NSP7 Interacts with Host Proteins

The interaction between viral and host proteins also affects viral replication and proliferation, which is a molecular base leading to pathogenic effects on the host body. NSP7 can interact with host RBM39 protein to promote PRRSV proliferation by down-regulating IFN-β production [30]. Dong et al. [44] identified 12 potential host proteins interacting with NSP7 using high-throughput proteomics techniques, including 60S ribosomal protein L34 (RL34), histone H2B type 1-J (H2B1J), heterogeneous nuclear ribonucleoproteins C1/C2 (HNRPC), peroxiredoxin (PRDX)2, PRDX4, PRDX1, D-3-phosphoglycerate dehydrogenase (SEAR), 60 kDa heat shock protein, mitochondrial (CH60), Nucleolin (NUCL), keratin, type II cytoskeletal 2 epidermal (K22E), tubulin alpha-1C chain (TBA1C), and actin, cytoplasmic 1 (ACTB), which may affect PRRSV proliferation and virulence by interacting with NSP7; the specific interactions between NSP7 and host proteins are shown in Figure 1. Through an inhibition experiment with PRDX4, Dong et al. [44] found that when PRDX4 function was inhibited, the titer of PRRSV progeny virus decreased; therefore, PRDX4 is considered to play an important role in the peroxidase-dependent antiviral immune response. PRDX1 and PRDX4 regulate the nuclear factor-κB (NF-κB) signaling pathway induced by a respiratory syncytial virus (RSV) and prevent RSV-induced oxidative damage of intermediate filaments and actin in epithelial cells [45]. The interaction between NSP7 and other proteins affects the growth, reproduction, and virulence of PRRSV, which will contribute to the design and development of new vaccines and therapeutic drugs for PRRSV.

## 9. Application of NSP7 in PRRSV Diagnosis

PRRSV NSP7 has high conservation and strong immunoreactivity. NSP7 can be expressed in bacteria as a soluble recombinant protein; it is a convenient target for enzyme-linked immunosorbent assay (ELISA) antigen preparation, and its homology between different strains is higher than that of NSP1 and NSP2 [15]; therefore, the established NSP7-ELISA method is suitable for PRRSV diagnosis. For example, Brown et al. [15] detected 469 of 470 positive samples by NSP7-ELISA; 215 PRRSV-1 samples and 255 PRRSV-2 samples were further distinguished, indicating that NSP7-ELISA has high sensitivity and specificity for PRRSV serological identification and in differentiating PRRSV-1 and PRRSV-2 strains. A total of 510 serum samples from 30 PRRSV-2 infected pigs were collected, and serological tests were performed on NSP1, NSP2, NSP4, NSP7, and NSP8. The results showed that the NSP7 antibody could be detected 14 days after inoculation, and the NSP7 antibody titer was still relatively high 126 days after inoculation. The NSP7 antibody persisted for more than 202 days, and in the first 126 days after inoculation, NSP7-ELISA results were highly correlated with IDEXX-ELISA results (r = 0.84) [15]. Brown et al. used the recombinant NSP7 as an antigen to determine the ability of NSP7 dual ELISA to detect serum antibody reactions from pigs infected with various genetically different field strains and validated a dual ELISA has high sensitivity and specificity for the simultaneous detection and differentiation of serum antibodies against PRRSV-1and PRRSV-2 [15]. Therefore, NSP7-ELISA can be used as an important tool for PRRSV detection.

A simple and rapid immunochromatographic test strip for PRRSV antibody detection was established using a colloidal gold-labeled recombinant NSP7 antigen probe. This method can accurately identify PRRSV antibodies in experimentally and field-infected pig serum, as well as immunized piglet serum, and no cross-reaction with antibodies against other pathogens were detected. However, this study could not distinguish between antibodies induced by vaccination and viral infection [46]. The GP5 protein is the main target protein for detecting the PRRSV antibody, which can detect the antibody titer in pigs after vaccination. NSP7 shows the highest conservation among PRRSV NSPs, with minimal differences among strains, and NSP7 protein is suitable as an antigen to distinguish antibodies caused by inactivated vaccine inoculation and virus infection [12]. Using recombinant proteins, ELISA detection conditions were optimized to successfully construct a GP5-NSP7-ELISA detection method [47], which can determine whether an antibody in infected pigs was induced by an inactivated vaccine or viral infection; this improves the effectiveness and accuracy of PRRS diagnosis. A fluorescent microsphere immunochromatographic test for the rapid, sensitive, and quantitative detection of PRRSV antibodies was established by labeling goat anti-pig immunoglobulin as the diagnostic antigen of recombinant NSP7 with N protein as the capture antigen [48]. This new strip detection technology combines the advantages of ELISA and colloidal gold test strip technology, which can detect antibodies quantitatively, more quickly, and with ultra-high sensitivity. Furthermore, a multiplex fluorescent microsphere immunoassay using N and NSP7 proteins as antigens covalently coupled to Luminex fluorescent microspheres can detect antibodies in serum and oral fluid [49]. Further studies on NSP7 will provide a new approach to the establishment of a fast-multiplexing detection method for PRRSV.

PRRSV often causes co-infection/secondary infection with bacteria or parasites. A multiplex fluorescent microsphere immunological detection method can simultaneously detect porcine circovirus 2 capsid protein, classical swine fever virus E2, and PRRSV-NSP7 IgG antibodies, which improved the differential diagnostic efficiency of PRRSV infection [50]. In addition, Ji et al. [51] developed a single assay for the detection of PRRSV NSP7 antibody, SIV HA antibody, and HEV ORF2 antibody on liquid-phase protein microarrays and a multi-liquid-phase protein multiplex assay for the detection of HEV, PRRSV, and SIV antibodies, providing a high-throughput, reproducible, and specific antibody detection system for HEV, PRRSV and SIV antibody monitoring. Huang et al. [52] designed and synthesized a pair of specific primers targeting the conserved region of NSP7 for PRRSV-2, optimized the reaction conditions, and established an SYBR green I fluorescence quantitative PCR detection method. This method showed high sensitivity and could quickly and accurately distinguish American strains from other viruses. However, SYBR green I can bind to increase all DNA double strands, the amplified primer dimer, single-strand secondary structure, and false amplification products, resulting in false-positive results. Therefore, this disadvantage can be reduced by designing appropriate primers, limiting the primer concentration, and increasing the annealing temperature to minimize the effect of this non-specific product. Therefore, the SYBR green I fluorescence quantitative PCR detection method has high PRRS diagnostic value in clinical practice.

Research shows that NSP7 monoclonal antibodies can recognize the same antigen epitope [53]. The recombinant NSP7 protein exhibited cross-immune reactions with antibodies against PRRSV-1 and PRRSV-2 by using the full-length *NSP7* gene for recombinant expression [54], suggesting that PRRSV-1 and PRRSV-2 have similar epitopes. The similar epitopes in NSP7 were truncated and expressed, showing no cross-immune reaction between the two genotypes, and the immune response activity was enhanced. This study thus established the foundation for the development of PRRSV diagnostic reagents. The high-affinity polypeptide of PRRSV NSP7 screened by phage screening technology has a similar effect with NSP7 monoclonal antibody and interferes with PRRSV replication [55]. Phages carrying NSP7 peptides have the potential to be used as PRRSV diagnostic reagents and antiviral drugs. The epitope distribution of the PRRSV NSP7 protein was also screened using phage display technology [56]. A new linear epitope, NAWGDEDRLN, was found on the PRRSV-2 NSP7β subunit at amino acids 153–162, which showed good reactivity with PRRSV-positive serum. These findings provide effective reagents for serological detection of PRRSV.

## 10. Conclusions

NSP7 is a long-term intermediate product of PRRSV replication, which regulates virus replication and proliferation. NSP7 can interact with viral proteins, such as interacting with the RdRp structure of NSP9, interacting with NSP9 and NSP12, and forming a transcription complex, which affects virus replication and proliferation. PRRSV NSP7 is highly conserved and immunoreactive and can be expressed as a soluble recombinant protein in bacteria, and can be used as a target for preparing antigens in ELISA.

## Figures and Tables

**Figure 1 animals-13-02269-f001:**
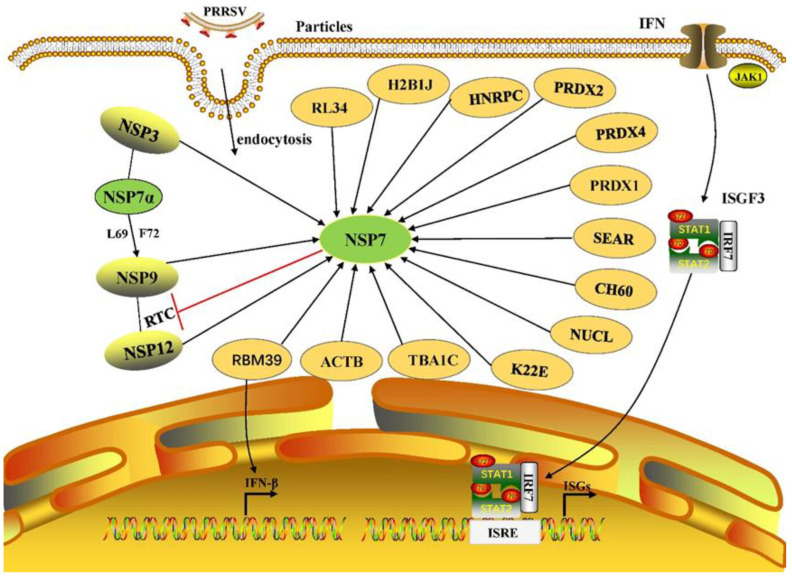
PRRSV NSP7 interactions with host and viral proteins. NSP5 has an indirect interaction with NSP9 through NSP7α, and the L69 and F72 amino acid residues of NSP7α play an important role in this process. NSP7 can interact with NSP9 and NSP12 to form an RTC. NSP7 can interact with host RBM39 protein to promote PRRSV proliferation by down-regulating IFN-β production. NSP7 inhibits IFN and ISGs expression by down-regulating IRF7 expression.

## Data Availability

All datasets are available in the main manuscript. The dataset supporting the conclusions of this article is included within the article.

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
