# Peer review of "Research Progress on Porcine Reproductive and Respiratory Syndrome Virus NSP7 Protein"

_animals, 2023, doi:10.3390/ani13142269_

Round 1

Reviewer 1 Report (Previous Reviewer 1)

Dear Author,

Many thanks for going back through my suggestions and making the appropriate amendments.

I have marked in yellow where I think the paper requires further clarification.

There are also several refences which I have been unable to find online- either PhD thesis or similar.  While the occasional reference to a PhD is satisfactory is should not be done extensively. There are also several refences which I cannot locate online and therefore I am unable to establish whether they are appropriate.  Please consider providing an alternative reference where necessary.

The English is adequate 

Author Response

Reviewer 2 Report (New Reviewer)

This is an interesting review manuscript aimed to study the biological characterization of NSP7 and its application in PRRS diagnosis, novel vaccine design, and therapeutic drug development. All topic subheadings included in this manuscript summarize very clear and well-supported information. I only suggest clarifying the purpose or main aim of the review in the Introduction section, and revise the structure of the Conclusions section. In addition, I recommend considering some minor grammar comments.

1)    General suggestions:

-       According to “Instructions for Authors”, the Introduction section should state the purpose or main aim of the manuscript.

-       Conclusions section looks like a long summary of the review. I suggest including only 2 or 3 conclusive sentences, and removing references as they should be included in the subheading topics.

-       In the References section, in the title of the reference only the first letter should be capitalized.

2)    Minor grammar comments:

-       Line 81: Replace “and” by “whereas”.

-       Line 83: Remove the word “also”.

-       Line 145: Separate the square bracket from the text.

-       Line 440: Page range is missing.

-       Line 443: Page range is missing.

-       Line 445: Remove period signs form the abbreviated Journal name.

-       Line 447: Page range is missing.

-       Line 473: Remove period signs form the abbreviated Journal name.

-       Line 483: Remove period signs form the abbreviated Journal name.

-       Line 500: Remove period signs form the abbreviated Journal name.

-       Line 503: Replace “Viruses-Basel” by “Viruses (Basel)”.

-       Line 503: Page range is missing.

-       Line 516: Page range is missing.

-       Line 556: Journal name should be abbreviated.

-       Line 561: Journal name should be abbreviated.

-       Line 574: Remove period signs form the abbreviated Journal name.

-       Line 580: Remove period signs form the abbreviated Journal name.

-       Line 582-583: Journal name should be abbreviated.

-       Line 591-592: Journal name should be abbreviated.

-       Line 592: Page range is missing.

-       Line 594: Journal name should be abbreviated.

Author Response

This manuscript is a resubmission of an earlier submission. The following is a list of the peer review reports and author responses from that submission.

Round 1

Reviewer 1 Report

This is a review summarising recent data describing the non-structural protein NSP7 of the PRRS virus.  The review focuses on the regions of sequence conservation and also describes the amino acid differences between strains.  There is also a description of some mutagenesis studies in attempt to better describe NSP7 function and how its modulate virus virulence and replication.  The paper reports that NSP7 is highly immunogenic and argues it may be a useful tool for PRRS diagnostics given the conservation of the protein between strains and the durability of the anti-NSP7 antibody following infection.

To summarise the article is not always well written and on occasions is quite unclear.  There are several paragraphs which have contradictory sentences. For example, in one sentence the author will state the NSP7 is highly conserved between strains, and then they will claim the protein is also highly variable.  I was also unable to find several of the references, some of which were PhD thesis and therefore I am unable to comment on their validity.  I would suggest that the authors try and include more peer-reviewed references in addition to PhD thesis, particularly where an important point is being made.  In addition, the conclusion doesn’t serve as a summary for the paper, for example, it doesn’t bring together any of the data relating to NSP7 function and how this may affect virus virulence and or attenuation despite the fact the authors spend a good proportion of the article reporting how NSP7 from different strains or with point mutations could explain differences in replication efficiency.  The authors should include this in their conclusion.   I suggest the authors take some time rewriting the conclusion and also consider what still remains to be done to identify the gaps in this area of research.

Line 33:  PRRSV is not a notifiable disease in England or the US, be specific.

Line 43:  ‘antigen type’ – incorrect term to use.  Can change this to ‘PRRSV is divided into two genetic species, according to differences in gene sequences and countries where the PRRSV species was first isolated.’

 Line 78:  remove ‘the best coincidence’ from the sentence. 

Line 81: It might be worth including that Chen was unable to detect NSP7Beta here.

Line 95: The author states that NSP7 shows good immunogenicity.  Would be good to provide a reference for this to back this statement up eg. Antibody Response to Porcine Reproductive and Respiratory Syndrome Virus (PRRSV) Nonstructural Proteins and Implications for Diagnostic Detection and Differentiation of PRRSV Types I and II | Clinical and Vaccine Immunology (asm.org)

Lines 112-117 :  The sentence ‘Despite the relative conservation of NSP7, the protein also shows certain variability’ is misleading.  The authors are referring to a paper (Zhang et al ) were site mutatagenesis was employed to try and understand key amino acid residues within NSP7 and how this might modulate virus replication of PRRSV.  The amino acids were determined by aligning PRRSV with EAV.  This is therefore not referring to natural variability of NSP7 between PRRSV strains,  However this sentence could be used before discussing the Li et al paper.  

Line 117:  I cannot find ref [17]- PhD thesis.

Line 125:  replace Marc 145 with MARC-145

Line 160- cannot find the paper referenced. [24]

Line 178:  cannot find [reference 10]

The results stated between lines 214-222 requires a conclusion with regards to NSP7.  The authors are referring to a study where a HP PRRS strain was isolated with various substitutions, some of which were included within the NSP7 but there was no change in virulence compared to the parent strains.  A conclusion sentence is required here.

Line 224-225:  the authors state that the design of an NSP7 tetramer contributes to the study of the host immune regulatory mechanisms.  It is not clear to me how this is the case and requires expansion or deleting, ‘further contributes to the study of NSP7 and the understanding of the host humoral response  to PRRS’ is a better sentence.

Line: 229-231 the authors state that NSP7 induces an antibody response and then go onto state in line 231 that NSP7 also induces a humoral response which is the process of antibody production which the authors have just discussed.   This therefore makes no sense and should be rewritten .

Line 230-232 :  the referenced paper [37] refers to NSP11 not NSP7.  If the author is intending to use NSP11 as an analogy to NSP7, this was not made clear, either do not refer to this paper or expand how this paper relates to the argument.

I am unable to find reference 39

Line 253: change INF to IFN

Line 323:  the following sentence doesn’t make sense: ‘However, when N  protein is used as an antiserum for physical examination’, I assume the authors are referring to antiserum raised again the N protein.  It is unclear what they mean by ‘physical examination’  I assume the authors are referring to the use of NSP7 antiserum in an ELISA format.  This is unclear and needs rewriting.

Line 326:  the authors state that the duration of the antibody response to N protein is shorter than that of NSP7 making differentiation between PRRSV1 and PRRSV2 difficult but this is not explained.  It is assumed this is due to less time for the antibody response to mature.  This needs to be rewritten.

Line 327:  The authors state that NSP7  shows potential for vaccine development, however, it  is well known that neutralising antibodies are a correlate of protection for PRRS, yet the authors give no evidence that NSP7 initiates neutralising antibody.  It is therefore unlikely that NSP7 would be a good vaccine candidate.  This statement should be removed.

Line 353:  the authors state that’ PRRSV infection most often occurs secondary to other pathogenic infection’.  This is wrong, and PRRSV more often causes co-infection/ secondary infection with bacteria or parasites.  Please rewrite this sentence accordingly.

Line 359:  the authors state:  the ‘correlation among them was significant’.  It is not clear what was correlating?  Do they authors mean that there was a strong correlation in the animals that tested positive for all 3 viruses discussed and the antibody titre?  Please elaborate.

Lines 365-370:  the authors describe the development of an RT-qPCR with high sensitivity and specificity but then claim that Syber-green can lead to false positives, which is confusing and misleading.  The authors should rewrite, be removing ‘high specificity’ and then add that that the ‘method maybe improved further through optimisation of the protocol’ as they describe in lines 368-370..

Line 393:  saying there is variability in NSP7 between strains is inconsistent with line 389 saying that NSP7 is highly conserved.  In line 393 it might be better to summarise the regions/ amino acids where there is variability between strains and give several examples.

Line 412 is incorrect:  Many studies have focused on the development of improved vaccines for PRRSV, although few studies have been successful.  Please alter the wording.

Line 413:  the generation of a humoral response doesn’t mean that the antibody is neutralizing.  Therefore, it is unlikely that NSP4, NSP7 and NSP12 would be useful targets for vaccine development unless they were able to induce neutralising antibody.  It would be better for the author to suggest that’ antibodies directed to NSP4, 7, and 12 require further characterisation,  to determine their utility in both PRRSV detection and vaccine development ‘.